# Molecular characterization and spatiotemporal expression of prohormone convertase 2 in the Pacific abalone, *Haliotis discus hannai*

**Md. Rajib Sharker**[1], **Ill-Sup Nou**[2], **Kang Hee Kho**[1]*

**1** Department of Fisheries Science, College of Fisheries and Ocean Sciences, Chonnam National University, Jeonnam, Republic of Korea, **2** Department of Horticulture, College of Life Science and Natural Resources, Sunchon National University, Jeollanam-do, Republic of Korea

* kkh@chonnam.ac.kr

**Data Availability Statement:** All relevant data are within the paper.

**Funding:** This research was a part of the project titled 'Development of technology for abalone

## Abstract

Prohormone convertases (PCs) are subtilisin-like proteases responsible for the intracellular processing of prohormones and proneuropeptides in vertebrates and invertebrates. The full-length PC2 cDNA sequence was cloned from pleuropedal ganglion of *Haliotis discus hannai*, consisted of 2254-bp with an open reading frame of 1989-bp and encoded a protein of 662 amino acid residues. The architecture of Hdh PC2 displayed key features of PCs, including a signal peptide, a pro-segment domain with sites for autocatalytic activation, a catalytic domain, and a pro-protein domain (P-domain). It shares the highest homology of its amino acid sequence with the PC2 from *H. asinina* and to lesser extent with that of *Homo sapiens* and *Rana catesbeiana* PC2. Sequence alignment analysis indicated that Hdh PC2 was highly conserved in the catalytic domain, including a catalytic triad of serine proteinases of the subtilisin family at positions Asp-195, His-236, and Ser-412. The cloned sequence contained a canonical integrin binding sequence, and four cysteine residues involved in the formation of an intramolecular disulfide link. Phylogenetic analysis revealed that the Hdh PC2 is robustly clustered with the Has PC2. Quantitative PCR assay demonstrated that the Hdh PC2 was predominantly expressed in the pleuropedal ganglion rather than in other examined tissues. Although PC2 mRNA was expressed throughout the gametogenetic cycle of male and female abalone, the expression level was significantly higher in the ripening stage of female abalone. Also, a significantly higher expression was observed in the pleuropedal ganglion and gonadal tissues at a higher effective accumulative temperature (1000˚C). *In situ* hybridization revealed that the PC2 mRNA expressing neurosecretory cells were distributed in the cortex region of the pleuropedal ganglion. According to the results, it can be concluded that pleuropedal ganglion is the highest site of PC2 activity, and this enzyme might be involved in the abalone reproduction process.

aquaculture using sperm cryopreservation' (Grant No. 2018-2129)' funded by the Ministry of Oceans and Fisheries, Korea.

**Competing interests:** All authors declare that they have no conflict of interest.

## Introduction

Prohormone convertases (PCs) are $Ca^{2+}$ dependent subtilisin-like endoproteases and are thought to be involved in the post-translational process of hormones, neuropeptides, and other regulatory proteins [1,2]. PCs play a pivotal role to convert an inactive neuropeptide precursor into an active mature peptide by limited proteolysis at multiple basic sites [3–5]. Several subtypes of PCs have been identified by molecular cloning and categorized as members of the subtilisin-like endoproteases family. These enzymes include furin, PC1/3, PC2, PC4, PACE4, PC5/6, PC7, SKI-1 (Mbtps1), and PC9 [6]. These neuropeptide proteases are structurally and functionally related to yeast Kex2, and are a homologue of bacterial subtilisin-like serine proteases [7,8]. The subtypes of PCs generally seem to be involved in the tissue-specific processing of multiple neuropeptide and peptide hormones. Some PCs (furin and PACE4) exhibited a ubiquitous tissue distribution, whereas the expression of others, including PC1 and PC2, is restricted to neural and endocrine cells [9]. Although each of these subtypes has distinct characteristics and specificities, similar biochemical properties are found among the members of PCs in both vertebrates and invertebrates [6]. The cDNA architecture of PCs contains an N-terminal signal peptide, a pro-peptide segment, a catalytic domain, a pro-protein domain (P-domain), and a carboxy terminal region with high sequence variability among different subtypes of PCs [10–12]. PC2 is responsible for the maturation of precursor molecules by endoproteolytic cleavage at pairs of basic amino acid residues in the regulated secretory pathway of neuroendocrine cells [13]. In *Xenopus*, PC2 plays a crucial role in the processing of proopiomelanocoretin (POMC) to α-MSH [13]. It is also involved in the processing of egg-laying hormone-related precursors in atrial-gland secretory cells of *Aplysia* [14]. Toullec et al.[15] reported that PC2 is the key endoprotease responsible for the maturation of crustacean hyperglycemic hormone (CHH). Homologues of PC2 have been characterized in only few invertebrates, including the nematode *Caenorhabditis elegans* [16], gastropod mollusk species *Lymnaea stagnalis* [17], *Aplysia californica* [14], *H. asinina* [18], arthropod species *Lucilia cuprina* [19], *Drosophila melanogaster* [20], *Orconectes limosus* [15], and *Penaeus monodon* [21].

The abalone is a marine gastropod species widely distributed throughout temperate and tropical coastal regions [22]. Of the *Haliotis* species, *H. discus hannai* is a highly valued seafood in the southern coasts of China, Japan, Taiwan, and Korea because of the presence of health-beneficial bioactive molecules [23]. Although many neuroendocrine hormones have been reported in *H. discus hannai*, the enzymes involved in post-translational modification of neuroendocrine hormones in Pacific abalone are lacking. Hence, the present study was conducted to isolate and molecularly characterize the PC2 in *H. discus hannai*.

## Materials and methods

### Experimental animals and sample collection

An adult female abalone (*H. discus hannai*) with shell length of 10.5 cm and total body weight of 148.2 g were collected from Naesan, Gogun-myeon, Jindo Island (34˚31'16.2"N 126˚22'28.7"E) and transferred to the laboratory in the Department of Fisheries Science, Chonnam National University. The animals were anesthetized with ethyl 3-aminobenzoate methane sulfonate (MS-222: 1g/L; Sigma-Aldrich, St. Louis, MO, USA) and the pleuropedal ganglion, cerebral ganglion, branchial ganglion, digestive gland, testis, ovary, gill, and mantle were collected. Each collected sample was frozen immediately in liquid nitrogen, and then stored at −80˚C for total RNA isolation.

For preparing the cryosection, the pleuropedal ganglion was washed in phosphate buffered saline (PBS; pH 7.4) and immersion fixed it in 4% paraformaldehyde (PFA) overnight. A brief

procedure of cryosection preparation from pleuropedal ganglion tissue was described by Sharker et al. [24].

Animal experiments were conducted in accordance with the guidelines of the Institutional Animal Care and Use Committee of Chonnam National University (CNU IACUC) and according to Article 14th of the Korean Animal Protection Law of the Korean government, and the animals were cared for in accordance with the Guidelines for Animal Experiments of Chonnam National University. No specific permissions are required to work with invertebrates in South Korea. Similarly, no permissions were needed for the collection of *H. discus hannai* from sample sites because they were not harvested from the protected area and this species is not an endangered or protected species.

## RNA isolation and cDNA synthesis

Total RNA was extracted from each tissue of Pacific abalone using an RNeasy mini kit (Qiagen, Hilden, Germany) and treated with RNase-free DNase (Promega, Madison, WI, USA) to eliminate the genomic DNA contamination. The concentration and integrity were then detected by spectrophotometry (NanoDrop® NP 1000 spectrophotometer) and electrophoresis on a 1% (w/v) agarose gel. Total RNA (1 μg) was reverse transcribed to cDNA using Superscript® III First-Strand synthesis kit (Invitrogen, Carlsbad, CA, USA) according to the manufacturer's protocol.

## Cloning and sequencing of PC2

In order to isolate and characterize PC2 cDNA, reverse transcription (RT) primers (sense: 5′- AGAGCTGGTCGTATGTAAGG -3′ and antisense: 5′- GCTACTCCTCCACTCTGTC -3′), were designed based on the *H. asinina* PC2 cDNA sequence (GenBank accession no. EU684323.1). PCR amplification was performed in a final reaction volume of 20 μL containing 1 μL (20 pmol) each of forward and reverse primers, 4 μL of 5× Phusion HF buffer (1×), 2 μL of dNTP (200 μM), 0.5 μL of 1 U Phusion DNA polymerase, 10.5 μL sterile distilled water (dH$_2$O), and 1 μL of the synthesized cDNA from the pleuropedal ganglion as a template. The cycling condition was as follows: 5 min at 94°C, followed by 35 cycles of 2 min at 94°C, 30 s at 58°C, 30 s at 72°C, with a final dissociation step of 5 min at 72°C. The amplified PCR products were separated on 1.2% agarose gel electrophoresis and purified using a Wizard SV gel and PCR clean-up kit (Promega). The purified PCR products were then ligated into the pTOP Blunt V2 vector (Enzynomics, Daejeon, Korea), and transformed into competent *E. coli* DH5α cells (Enzynomics). Plasmid DNA was extracted from the positive clones with a plasmid miniprep kit (Qiagen) and sequenced using the Macrogen Online Sequencing System (Macrogen, Seoul, Korea). Rapid amplification of 5′ and 3′ cDNA ends (RACE) were performed with a SMARTer® RACE 5′/3′ Kit (Clontech Laboratories, Inc., USA) according to the manufacturer's recommendation. Gene-specific primer sequences (GSPs), including a 15-bp overlap with the 5′-end of the GSP sequence (antisense primer: 5′-GATTACGCCAAGCTTGCTGGTCCAG CATTCTCAAGTCTGCAAC-3′, sense primer: 5′- GATTACGCCAAGCTTGTTGCAGACTTG AGAATGCTGGACCAGC-3′), a universal primer mix (UPM): 5′-CTAATACGACTCACTAT AGGGCAAGCAGTGGTATCAACGCAGAGT-3′, and SeqAmp DNA Polymerase in a final volume of 50 μL were used to conduct the RACE PCRs. Touchdown PCR was performed with 25 cycles for 3′-RACE and 30 cycles for 5′-RACE PCR following the kit instructions. Purification of RACE PCR products was done using NucleoSpin Gel and PCR Clean-Up kit and ligated them into the linearized pRACE vector, transformed them into Stellar Competent Cells supplied with the kit, and then sequenced them with the Macrogen Online Sequencing System (Macrogen, Seoul, Korea). Finally, the sequenced RACE products were assembled by overlapping with the initial fragment.

## Sequence analysis

To analyze the PC2 protein sequence of *H. discus hannai*, multiple online software programs were used. Basic Local Alignment Search Tool (BLASTP) (http://www.ncbi.nlm.nih.gov/BLAST/) was used to identify the protein homology of PC2 protein with the PC2 of other species. Predictions of N-linked glycosylation motifs and serine/threonine phosphorylation sites were performed with the NetNGlyc 1.0 server and NetPhosK 1.0 server, respectively. SignalP 4.1 (www.cbs.dtu.dk/services/SignalP/) was used to infer the N-terminal signal peptide, and the bonding state of cysteines in the protein sequence was determined using CYSPRED [25]. Physical and chemical properties associated with the primary sequence of the protein were calculated using ProtParam (http://expasy.org/tools/protparam.html), and subcellular localization was determined with Protcomp (http://www.softberry.com/berry.phtml). Multiple alignments of the deduced amino acid sequences of PC2 protein were accomplished with Clustal Omega [26,27]. Jalview, version 2.10.0 (www.jalview.org) was used for editing and visualizing the aligned sequence [28].

## Phylogenetic analysis

To construct a phylogenetic tree, PC2 protein sequences from invertebrates and vertebrates were retrieved from the NCBI database using the BLASTP algorithm. The sequences were aligned using Clustal Omega [26,27]. The tree was generated with MEGA software (version 6.06) using a neighbor-joining method with 1,000 bootstrapping replicates [29].

## Quantitative PCR (qPCR) analysis

The tissue expression pattern of PC2 mRNA was analyzed by qPCR assay using the 2× qPCRBIO SyGreen Mix Lo-Rox kit (PCR Biosystems, Ltd., London, UK) according to the manufacturer's protocol. Gene-specific primers (forward: 5′–ATGTAAGGAGGTCGAAGTGC–3′ and reverse: 5′–GTCTAGTATGTGGTACGCTTC–3′) and primers (forward: 5′– TGTCCGTTTCA CCAACAAGG–3′ and reverse: 5′– AGATGGAATCAAGTTTCAATT –3′) from ribosomal protein L-5 gene (RPL-5, JX002679.1) of *H. discus hannai* were used to normalize the expression of the target gene. The 20 μL reaction mixtures comprised 1 μL cDNA template of each tissue, 1 μL (10 pmol) of each forward and reverse primer, 10 μL SyGreen Mix, and 7 μL PCR-grade water. PCR was performed under the following conditions: pre-incubation at 94°C for 5 min, followed by three-step amplification at 95°C for 2 min, 60°C for 30 s, and 72°C for 30 s for 40 cycles. Three replicates (N = 3) were done for each qPCR product. The relative gene expression was analyzed on the basis of the $2^{-\Delta\Delta ct}$ method [30].

## Expression of PC2 mRNA in gonads during gametogenesis

qPCR assay was performed to assess the expression levels of PC2 mRNA in the gonads at different stages. The stages of the gonad were classified according to a previous study [31]. qPCR assay and analysis of relative mRNA levels were performed as previously described.

## Expression of PC2 mRNA in neural ganglia and gonad at Effective Accumulated Temperature (EAT)

The mature abalones were obtained from the hatchery and kept in the tanks with filtered seawater and continuous aeration at 9.5°C for one month. The sample preparation and expression level of PC2 mRNA transcript at different EAT was found according to the method described by [31]. One microliter of cDNA template from the neural ganglia and gonadal tissues at different EAT were used to do the qPCR assay.

## *In Situ* Hybridization (ISH)

DIG-labeled antisense and sense RNA probes were prepared from the coding region of the PC2 nucleotide sequence by *in vitro* transcription following previous studies in *H. discus hannai* [24,31]. The pleuropedal ganglion tissue sections were pre-hybridized with hybridization buffer and yeast total RNA (50 μL) for 2 h, followed by overnight hybridization with the RNA probe at 65˚C. The hybridized tissue sections were sequentially washed, and then non-specific binding was blocked with 10% calf serum for 1 h at room temperature. The sections were incubated at 4˚C overnight with alkaline phosphatase-conjugated anti-digoxigenin-Ap-Fab fragments antibody (diluted 1:500 in blocking solution [Roche]) to detect the hybridization signal. Finally, the sections were treated with the labeling mix (2 ml alkaline tris buffer, 9 μl nitroblue tetrazolium, 7 μl 5-bromo-4-chloro-3-indolyl phosphate disodium salt) and incubated in a dark and humid chamber for at least 1 h to observe the color. After optimal color development, the sections were observed and photographed using a stereo microscope (SMZ1500, Nikon, Tokyo, Japan).

## Nuclear fast red counterstain

Antisense probe hybridized ISH slides were counterstained using nuclear fast red (Sigma-Aldrich, USA). Slides were rinsed with distilled water and then incubated in nuclear fast red solution for 5 min followed by washing in tap water for 3 min. Slides were dehydrated using ascending series of ethanol, dipped in histo-clear (National diagnostics, USA) for 3 min, and finally cover-slipped with permount mounting medium.

## Statistical analysis

Data were analyzed using one-way analysis of variance (ANOVA), followed by Tukey's multiple comparisons using SPSS (version 16.0) to detect the differences in relative mRNA expression levels. All the data are presented as mean ± SD, and a difference of $p < 0.05$ was regarded as statistically significant.

## Results

### Cloning and characterization of PC2 from Pacific abalone

The full-length PC2 cDNA sequence was obtained from the pleuropedal ganglion and referred to as Hdh PC2. The sequence data have been deposited in the GenBank database under the accession number MN822082. A total of 2254-bp cDNA transcript of PC2 contained a 143-bp 5′-untranslated region (UTR) and a 122-bp 3′-UTR with a canonical polyadenylation signal sequence (AATAAA) located 13-bp upstream of the poly-A tail. The open reading frame showed a complete coding sequence of 1989-bp that encodes a putative protein of 662 amino acids with a theoretical molecular mass of 73.38 kDa and an isoelectric point of 6.91 (Fig 1).

*In silico* analysis (protcomp, http://www.softberry.com/berry.phtml) indicated that the subcellular localization of this deduced protein is in the membrane bound golgi network. The cloned sequence contained a 29-amino acid $NH_2$-terminal signal peptide, a pro-segment region, a catalytic domain, and a pro-protein domain (P-domain) with a variable C-terminal region. The three active sites residue ($D^{195}$, $H^{236}$, $S^{412}$) consisting of a catalytic triad of serine protinases of the subtilisin family were found in this sequence. The $ASP^{338}$ residue was believed to be involved in the oxyanion stabilization. A canonical integrin binding signatures (RGD) were present in the P-domain of Pacific abalone PC2. Three potential N-linked glycosylation motifs (Asn-281, Asn-311, and Asn-403), and four cysteine residues (Cys-240, Cys-253, Cys-347, and Cys-498), were identified which could potentially form an intrachain

```
1      TTTTTGCCAAGAACGACCAGGTCGGGGATTCAAGCAGGGGAAAGGAACATCAGATATTGA

61     AGTAGAAACATCGAAAACAACACCTGAAGATTGTTGTTGGAAACAGAAAAGGAGAAACCG

121    GAATTGAGGGCGTACTGTTTGAAATGGTTAATTTTCAGTCATGGAGAACTAGACGATTTC
                                   M  V  N  F  Q  S  W  R  T  R  R  F

181    TGGGAATATTGGTAACATTGGCTCTGGTCCTTCCCGAGCTGGTCGTATGTAAGGAGGTCG
       L  G  I  L  V  T  L  A  L  V  L  P  E  L  V  V  C  K  E  V

241    AAGTGCTTACTAACTCCTGGTTAGTGGAGCTTGAGGCCCCCGGAGGAGCAAGGGTCGCCA
       E  V  L  T  N  S  W  L  V  E  L  E  A  P  G  G  A  R  V  A

301    GAGATGTAGCAAAGAGAACAGGATTTACCTATGTGTCTCCGGTCCTCAACTCACAAACAC
       R  D  V  A  K  R  T  G  F  T  Y  V  S  P  V  L  N  S  Q  T

361    AGCTTCATCTCATCCATAAGGGCGTGCACCATGCCAGATCAAAGAGAAGCGTACCACATA
       Q  L  H  L  I  H  K  G  V  H  H  A  R  S  K  R  S  V  P  H

421    CTAGACTTTTAAAGGCCCATCCCTATGTAAAAAGCGCCGTTCAACTCACTGGATATTTAC
       T  R  L  L  K  A  H  P  Y  V  V  K  S  A  V  Q  L  T  G  Y  L

481    GTCAGAAAAGGGGTTATAAATCTTTGGATGCTATACTAACGGATTTTAGGCTACAAAAGC
       R  Q  K  R  G  Y  K  S  L  D  A  I  L  T  D  F  R  L  Q  K

541    CCAAAATAGTAAATTACCCAGTTTTTGAAAGGTCAAAGCCAAAGTTACCCTCAGATCCAG
       P  K  I  V  N  Y  P  V  F  E  R  S  K  P  K  L  P  S  D  P

601    ATTTTGACAAAGAAGGATATTTGAGGAACACGGGACAAAGCGGCGGAGTGGCCGGCCTGG
       D  F  D  K  E  G  Y  L  R  N  T  G  Q  S  G  G  V  A  G  L

661    ACCTGAACGTGGTGGAAGCATGGGAGATGGGCTACACCGGAACTGACGTCACCACCGCTA
       D  L  N  V  V  E  A  W  E  M  G  Y  T  G  T  D  V  T  T  A

721    TAATGGATGATGGTATCGACTATCTGCACCCCGACCTGCGACACAACTACAACGCCGAAC
       I  M  D  D  G  I  D  Y  L  H  P  D  L  R  H  N  Y  N  A  E

781    ATAGTTATGATTTCAGCAGCAACGACCCATACCCATACCCCCGCTATACAGACACATGGT
       H  S  Y  D  F  S  S  N  D  P  Y  P  Y  P  R  Y  T  D  T  W

841    TCAATAGTCATGGGACTCGATGCGCTGGCGAAGTATCCGCCGCCAGAGATAAAGGTAATT
       F  N  S  H  G  T  R  A  G  E  V  S  A  A  R  D  K  G  N

901    GTGGAGTAGGCGTGGCTTACGGGTCAAAGGTTGCAGACTTGAGAATGCTGGACCAGCCGT
       G  V  G  V  A  Y  G  S  K  V  A  D  L  R  M  L  D  Q  P

961    TCATGACAGACCTGATAGAGGCCAACCTAACCGGCCACATGTCTAACTTGCCATACATCT
       F  M  T  D  L  I  E  A  N  L  T  G  H  M  S  N  L  P  Y  I

1021   ATAGCGCTAGCTGGGGACCCACAGACGACGGCAAGACGGTTGATGGCCGCAGGAACTTAT
       Y  S  A  S  W  G  P  T  D  D  G  K  T  V  D  G  R  R  N  L

1081   CAATGAGGGCCATCGTCAACGGCGTCAATAACGGTCGTAATGGCAAGGGCAACATCTACG
       S  M  R  A  I  V  N  G  V  N  N  G  R  N  G  K  G  N  I  Y

1141   TCTGGGCGTCCGGCGACGGCGGCCCCAATGACGACTGCAACTGTGACGGCTACGCTGCCA
       V  W  A  S  G  D  G  G  P  N  D  D  C  N  D  G  Y  A  A

1201   GTATGTGGACCATATCCATCAACTCAGCCACCAACGACGGCCAGACTGCCGGCTATGACC
       S  M  W  T  I  S  I  N  S  A  T  N  D  G  Q  T  A  G  Y  D

1261   TCAGCTGCTCCTCCACTCTCGCCTCCACCTTCAGCAACGGCAAAGCAACGTCACGAGATG
       L  S  C  S  S  T  L  A  S  T  F  S  N  G  K  A  T  S  R  D

1321   CTGGCGTGGCCACCACCGACCTGTACGGTAACTGCACCGCCAGCCATTCCGGGACATCAG
       A  G  V  A  T  T  D  L  Y  G  N  C  T  A  S  H  S  G  T  S

1381   CAGCTGCCCCCGAGGCATATGGAGTTTCGCTCTAGCCCTTGAAGCAAATCGAAACCTGA
       A  A  A  P  E  A  Y  G  V  F  A  L  A  L  E  A  N  R  N  L

1441   ACTGGCGTGAGATTCAGCACCTGACTGGTCTCACCTCCAAGAGGAACTCACTCTCCGACT
       N  W  R  E  I  Q  H  L  T  G  L  T  S  K  R  N  S  L  S  D

1501   CCAACGGCGTCCACGCATGGAAGTATAGCGGCGCGCCGGTCTGGAGTTCAACCACCTCGTCG
       S  N  G  V  H  A  W  K  Y  S  G  A  G  L  E  F  N  H  L  V

1561   GCTACGGCGTGCTTGACGCTGCCCTGATGGTGGATCTGGCCCATACTTGGAAGGGGCTTC
       G  Y  G  V  L  D  A  A  L  M  V  D  L  A  H  T  W  K  G  L

1621   CCGAACGTTTCCATTGTACCGCCGGCTCTGACAGCACCGAGAGGTATTTCGACATGTGCA
       P  E  R  F  H  T  A  G  S  D  S  T  E  R  Y  F  D  M  C

1681   ACCCCATCCGTATCACCATTGACACCGACGGTTGTGCCGGAAGCGATAGCGAAGTCAACT
       N  P  I  R  I  T  I  D  T  D  G  C  A  G  S  D  S  E  V  N

1741   ACTTGGAGCATGTGCAGTCATTTATCACTCTGCGAGCAACGTTCCGAGGAGACGTCACCA
       Y  L  E  H  V  Q  S  F  I  T  L  R  A  T  F  R  G  D  V  T

1801   TCTACCTCACGTCTCCCATGAGTGATACATCGATGACGTTGAGCCAGCGACCCAACGAAG
       I  Y  L  T  S  P  M  S  D  T  S  M  T  L  S  Q  R  P  N  E

1861   ACGACGGCAAGAACAGCACTACCCGGTGGCCCTTCATGACCGTACACACCTGGGCCGAGT
       D  D  G  K  N  S  T  T  R  W  P  F  M  T  V  H  T  W  A  E

1921   TGTCACATGGTCGCTGGACCCTTGAAGTTATCCTAGAACCAATCCGTGGTCTGAAGACAA
       L  S  H  G  R  W  T  L  E  V  I  L  E  P  I  R  G  L  K  T

1981   ACTATGAGGAAGGAGTCTTCAAGGAATGGACCCTGGTCATGCATGGATCGAAGGAACAAC
       N  Y  E  E  G  V  F  K  E  W  T  L  V  M  H  G  S  K  E  Q

2041   CCTACAGGGACCAGCCTGATAACAAGGCCAAACATCAGAAGCTGTACAAGCCCAAGCGAC
       P  Y  R  D  Q  P  D  N  K  A  K  H  Q  K  L  Y  K  P  K  R

2101   AGCATGCTAGTGGCATTGCCTTCCGGAAATAACCCCACCCGGGGGCGAGAGAGCCCCCGG
       Q  H  A  S  G  I  A  F  R  K  *

2161   GGCCGGGGACCCTCCAAGGTTTCCTGGAAACAAAACTTCGGTTTGAAATAAAATATTATC

2221   GAATCAAAAAAAAAAAAAAAAAAAAAAAAAAAAAAA
```

**Fig 1. The nucleotide and deduced amino-acid sequences encoding a PC2 of Pacific abalone.** The initiation and termination codon (asterisks), and the classical polyadenylation signal are in the bold font. The N-terminal signal peptide is underlined with dots. The potential N-linked glycosylation and phosphorylation sites are shown by circles and triangles, respectively. Four cysteine residues (Cys-240, Cys-253, CYS-347, and Cys-498) that form a disulfide link are shaded in green. The active-site residues (ASP, His, and Ser) are circled, and the ASP residue stabilizing the oxyanion is boxed. The canonical integrin-binding motif is denoted by a green background. The pro-region, catalytic site, and P-domain are delimited by a broken line with arrows, a solid line, and a broken line, respectively.

disulfide link. The threonine and serine residues at positions $166^S$, $231^T$, $377^S$, $381^S$, $444^T$, $462^S$, and $505^T$ serve as potential sites for phosphorylation by protein kinase A or C. A BLASTP search indicated that this translated protein sequence showed the highest homology (95%) with the *H. asinina* PC2 (Has PC2). It also exhibited 73%, 71%, 66%, and 61% identity with *A. californica*, *L. stagnalis*, *Mizuhopecten yessoensis*, and *Limulus polyphemus* PC2, respectively.

Multiple sequence alignment analysis demonstrated that the potential cleavage site indicating the pro-segment region was conserved remarkably among different invertebrate species. The catalytic triad and oxyanion hole residues were also well conserved in the PC2 sequences (Fig 2).

The integrin binding sequence $Arg^{547}$ and $Gly^{548}$ were conserved in all organisms, whereas the $ASP^{549}$ residue was replaced by a Cys residue in *P. canaliculata*, *L. stagnalis*, and *A. californica*, and a Tyr residue in *H. asinina* and *H. rubra*.

The constructed phylogenetic tree revealed several distinct clades. Hdh PC2 is contained in the gastropod PC2 clade and is more closely related to Has PC2. The *H. discus hannai* serine protease was used as a outgroup (Fig 3).

qPCR assay was performed to investigate the mRNA expression profile of Hdh PC2 in neural ganglia (cerebral ganglion, branchial ganglion, pleuropedal ganglion), digestive gland,

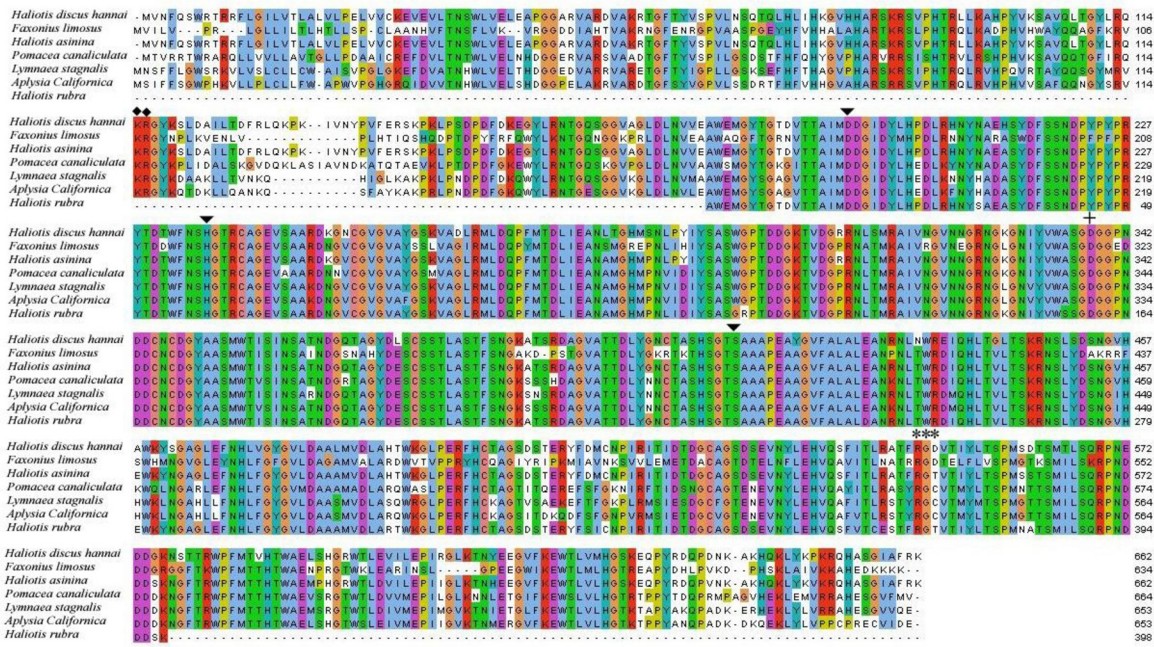

**Fig 2. Comparative alignment of the predicted Hdh PC2 amino-acid sequences of the Pacific abalone with those of other invertebrate PC2.** The pro-domain indicating the cleavage site is denoted by a diamond circle. The catalytic triad active-site residues and the cognate integrin-binding residue are indicated by black arrowheads and asterisks, respectively. Plus sign (+) represents the oxyanion hole residue.

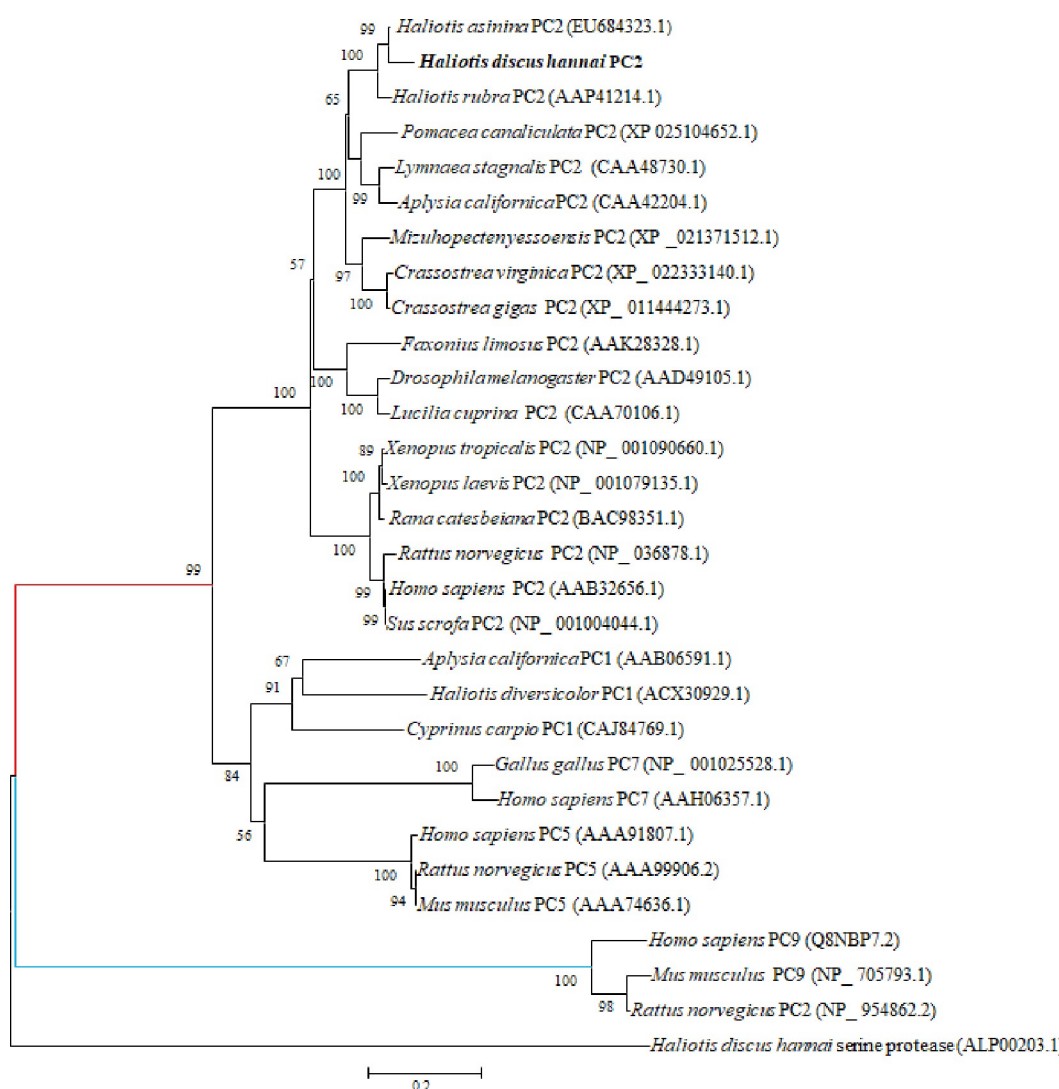

**Fig 3. Molecular phylogenetic analysis of Hdh PC2 using a neighbor-joining method after Clustal Omega alignment.**
The 1000 bootstrap replicates to construct the phylogenetic tree. The 0.20 scale bar indicates the number of amino acid
substitutions per site. The Hdh PC2 in this study is highlighted in the bold font.

gonad (testis, ovary), gill, and mantle using gene-specific and ribosomal protein L-5 (RPL-5)
primers. The RPL-5 gene (JX002679.1) of *H. discus hannai* was used as internal control based
on its expression stability. The results of the qPCR showed that the relative mRNA abundance
of Hdh PC2 was significantly ($p < 0.05$) higher in the pleuropedal ganglion than in other
examined tissues (Fig 4).

The Hdh PC2 mRNA was expressed differently in the gonad at different gametogenesis
stages of Pacific abalone. In female, PC2 mRNA was expressed moderately in the degenerative,
spent, and active stages, but a significantly higher level of expression was found in the ripening
stage (Fig 5).

Although the relative mRNA expression level showed a rising tendency in the male repro-
ductive cycle, there were no statistical differences between the different gametogenetic stages.

The expression levels of Hdh PC2 mRNA in the neural ganglia and gonad was investigated
by qPCR at different effective accumulative temperatures (EAT). It has been shown that the

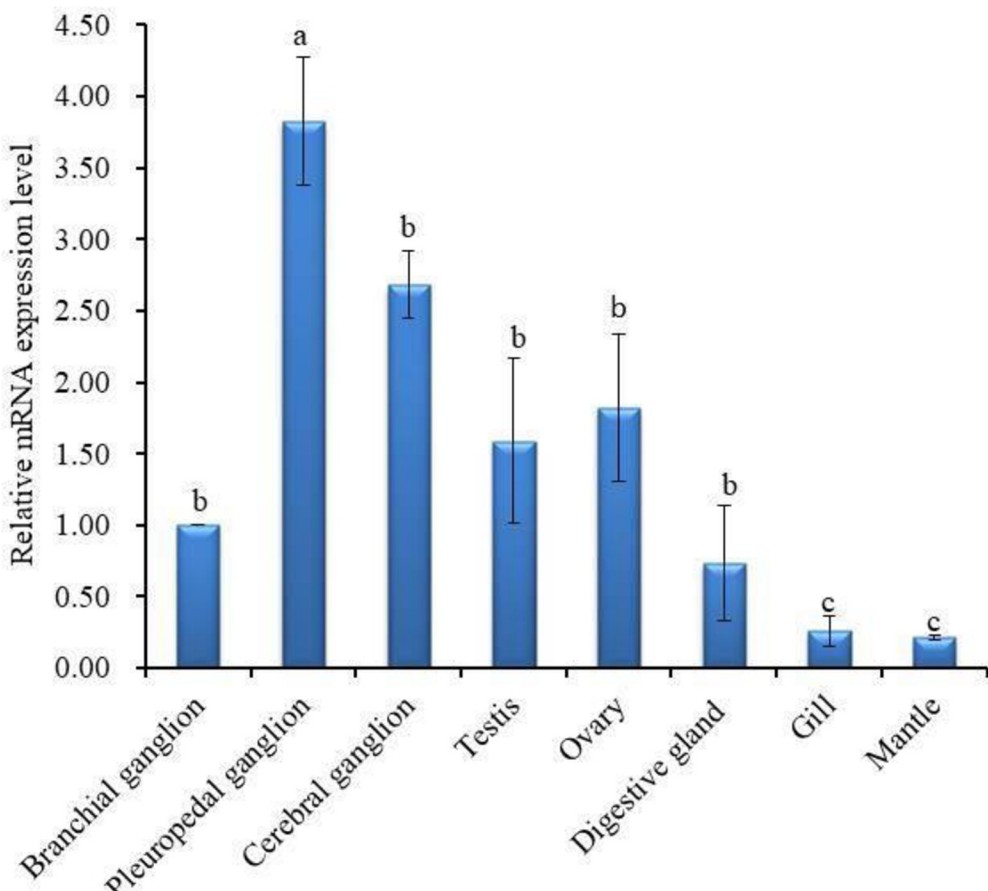

**Fig 4. Different mRNA expression levels of Hdh PC2 (means ± SD, $N$ = 3) in various tissues were quantified using qPCR.** Data were compared with the value of the branchial ganglion, which was assigned a relative value of 1. Different letters indicate significant differences ($p < 0.05$).

pleuropedal ganglion, testis, and ovary exhibited significantly higher expression at 1000˚C (Fig 6). There were no significant differences observed in other ganglia at different EAT.

The site of PC2 mRNA expression in the pleuropedal ganglion section was found by *in situ* hybridization with a DIG-labeled antisense probe. The positive hybridization signal for PC2 mRNA transcript was found in cells shown in purple color (Fig 7A, 7B, 7C and 7D).

These types of signals were not visualized in the negative control section, because of the absence of an antisense probe in the hybridization mix during incubation (Fig 7E). The results of fast red counterstaining indicated that the positive signal was likely localized in the neurosecretory cells of the cortex region (Fig 7F).

## Discussion

Prohormone convertases act as a molecular switch for the processing of biologically inactive polypeptide precursors to active peptides by limited endoproteolysis. This proteinase determines the cell type and time at which mature products are derived from a given inactive precursor protein, thereby profoundly affecting cellular communication, differentiation, and metabolic activity [9]. In mollusks, the PC2 was first reported in the cerebral ganglion of the central nervous system of the freshwater snail, *L. stagnalis* [17]. To date, reports on the

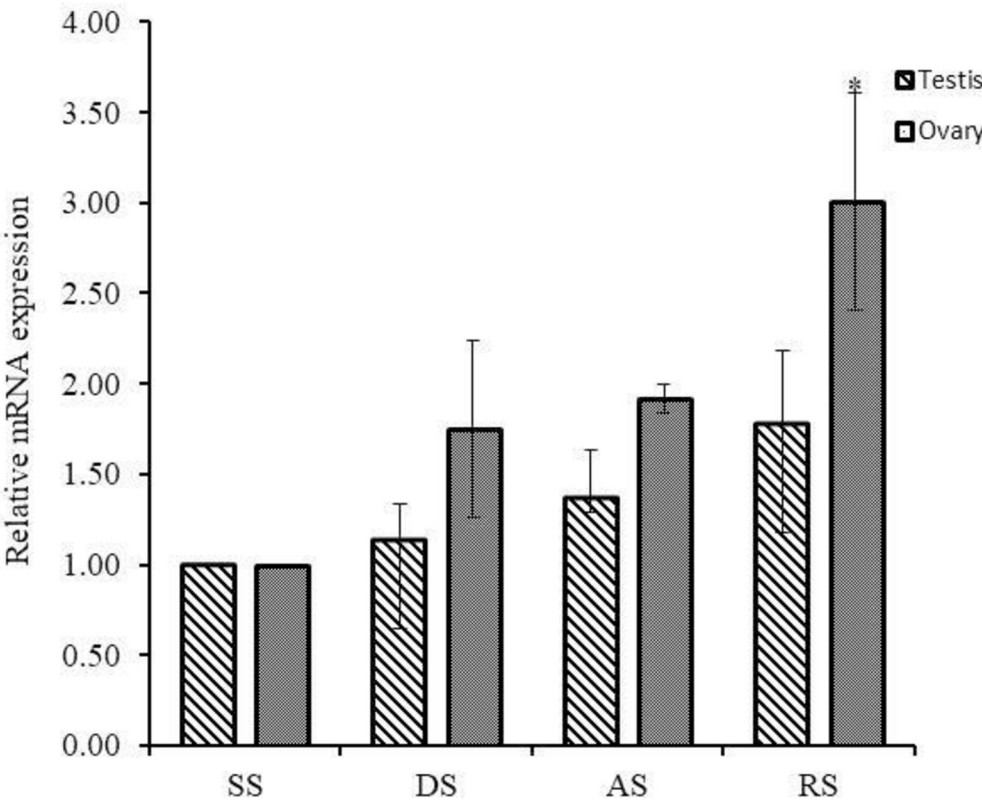

**Fig 5. Quantitative PCR analysis of Hdh PC2 mRNA expression in the gonads (ovary and testis) at different gametogenetic stages.** Asterisks indicate significant differences ($p < 0.05$). DS, Degenerative stage; AS, Active stage; RS, Ripening stage; SS, Spent stage.

molecular characterization and expression profile of PC2 in *H. discus hannai* has been lacking. For the first time, this study demonstrated the mRNA sequence encoding the PC2 from the pleuropedal ganglion of *H. discus hannai*. Similar to other PCs, the structural profile of Hdh PC2 possesses several recognition motifs, including a hydrophobic signal peptide, a pro-segment, a subtilisin-homologous catalytic region, a P-domain, and a variable C-terminal region (Fig 1). The catalytic domain contained Asp, His, and Ser active site residues, which seem to be involved in transition-state stabilization [32]. The catalytically important residue Asp[338], which is thought to be important for oxyanion stabilization during catalysis [33] as well as for interaction with the specific binding neuroendocrine polypeptide 7B2 in endoplasmic reticulum [34,35]. The presence of tyrosine sulfation, N-linked glycosylation, and phosphorylation motif might be crucial for preventing PC2 degradation in ER [36]. In the P-domain, the integrin-recognition sequence (RGD) plays a critical role in the intracellular sorting of enzymes into secretory granules as well as for controlling the stability of the enzyme within the ER [37]. Three putative N-linked glycosylation motifs were evident in Hdh PC2 (Fig 1), predicting it to be a glycoprotein. The predicted protein might be localized in the membrane bound golgi network which is in agreement with the results of previous studies [6].

The outcome of multiple sequence alignment indicated that the Hdh PC2 showed high degree of sequence identity in the catalytic region, proving that these residues are crucial in the catalytic activity of the enzyme (Fig 2). This result is consistent with previous reports [18,38].

The constructed phylogenetic tree revealed that the Hdh PC2 is robustly clustered with the Has PC2 and is most similar to other molluscan PC2s (Fig 3). Similar reports have been

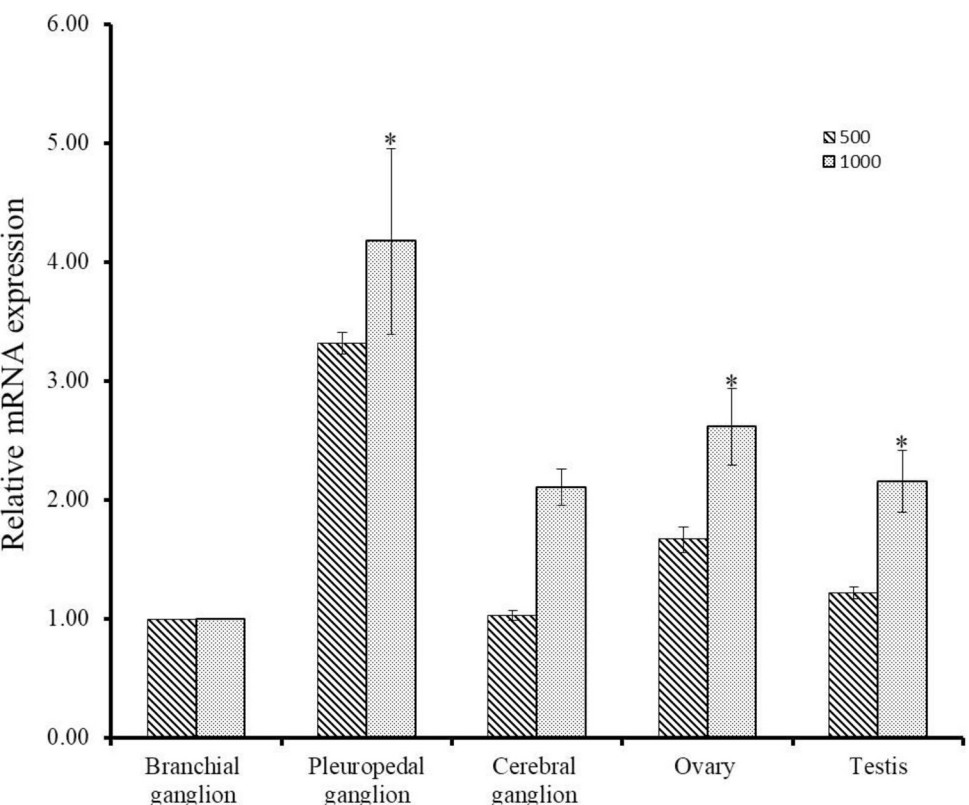

**Fig 6. Expression pattern of Hdh PC2 mRNA in neural ganglia and gonads at different Effective Accumulative Temperatures (EAT).** The mRNA levels were quantified by qPCR. Asterisks indicate significant differences ($p < 0.05$).

published by Cummins et al. [18], who observed that the PC2 of *H. asinina* is most closely related to *H. rubra*, *L. stagnalis*, and *A. californica* homologues.

The tissue-specific expression and relative mRNA expression of Hdh PC2 were determined using a qPCR assay. In the present study, a significantly higher expression was found in the pleuropedal ganglion than in other examined tissues (Fig 4). These results suggest that pleuropedal ganglion could be the main site of PC2 activity in abalone.

In order to explore the physiological activities of PC2 in *H. discus hannai* reproduction, the expression profile of PC2 mRNA was analyzed at different phases of the reproductive cycle. The results showed that the expression level of PC2 is higher in the ripening stage, suggesting that PC2 might be involved in reproductive regulation of the Pacific abalone. The level of a follicle-stimulating hormone (FSH) and luteinizing hormone (LH) would increase via the conversion of a prohormone into a mature peptide during the reproductive season of the freshwater snail [39]. Liu and Sun [40] reported that the egg-laying hormone usually existing as a biologically inactive precursor, needed to be converted into an active form by protease cleavage, such as prohormone convertase. PCs may play an important role in the processing of gonadotropin-releasing hormone, which is essential for the maturation of the gonads [41].

In abalone, effective accumulative temperature (EAT) is an influential factor for the regulation of gonadal maturation and spawning [42]. To date, no reports on the PC2 mRNA expression in tissues at different EAT in the abalone species have been published. In this study, the mRNA abundance of PC2 was highest in pleuropedal ganglion and gonadal tissues at higher

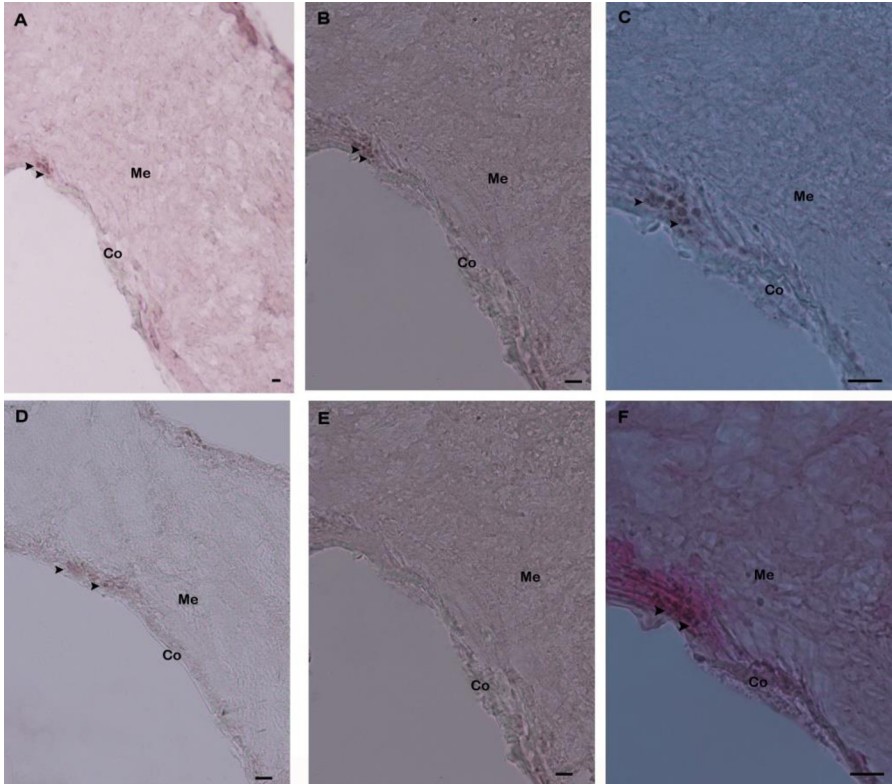

**Fig 7. *In situ* detection of Hdh PC2 mRNA in the pleuropedal ganglion of Pacific abalone. (A)** Positively stained neurosecretory cells expressing the Hdh PC2 mRNA in the cortex region is indicated by arrowheads; **(B)** Medium magnification of A; **(C)** High magnification showing hybridized Hdh PC2 mRNA in neurosecretory cells; **(D)** Medium magnification of the section showing a positive hybridization signal in the other part of the cortex region; **(E)** Hybridized with the Hdh PC2 sense riboprobe showing no hybridization signal; **(F)** Fast red counterstaining of section C demonstrated hybridized neurosecretory cells. Co, Cortex; Me, Medullae. Scale bars, 100 μm.

EAT (Fig 6). The results suggest that the rate of gonadal development and quantity of gametes increases with increasing EAT.

Previously, the distribution and expression of PC2 has been studied in the clawed frog [13], bullfrog [43], and medaka [38] by using *in situ* hybridization. An *in situ* hybridization experiment with the antisense mRNA in *H. asinina* PC2 revealed that PC2 mRNA transcripts were present in the cerebral and pleuropedal ganglia [18]. The *Lymnaea* PC2 mRNA was predominantly expressed in the central nervous system [17]. In this study, *in situ* hybridization of PC2 was shown to be expressed in the neurosecretory cells of the pleuropedal ganglion. All these data suggest that the PC2 enzyme might be synthesized in neural ganglia and be essential for intracellular processing of the prohormone that is involved in the gonadal maturation of abalone.

## Supporting information

**S1 Raw Images.**
(PDF)

## Author Contributions

**Conceptualization:** Md. Rajib Sharker, Kang Hee Kho.

**Data curation:** Md. Rajib Sharker.

**Formal analysis:** Md. Rajib Sharker.

**Funding acquisition:** Kang Hee Kho.

**Methodology:** Md. Rajib Sharker.

**Supervision:** Kang Hee Kho.

**Validation:** Md. Rajib Sharker.

**Writing – original draft:** Md. Rajib Sharker, Kang Hee Kho.

**Writing – review & editing:** Ill-Sup Nou.

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
