## [Decision Letter · Decision Letter 0]

27 Feb 2020

PONE-D-20-03317

Molecular characterization and spatiotemporal expression of prohormone convertase 2 in the Pacific abalone, Haliotis discus hannai

PLOS ONE

Dear Dr Kang Hee Kho,

Thank you for submitting your manuscript to PLOS ONE. Despite the difficulties in getting a second reviewer, the reviewer 1 has performed a very detail and constructive evaluation to which I fully aggre. After careful consideration, we both feel that it has merit but does not fully meet PLOS ONE’s publication criteria as it currently stands. Therefore, we invite you to submit a revised version of the manuscript that convincingly addresses all the points indicated by the reviewer.

We would appreciate receiving your revised manuscript by April 1st. To enhance the reproducibility of your results, we recommend that if applicable you deposit your laboratory protocols in protocols.io, where a protocol can be assigned its own identifier (DOI) such that it can be cited independently in the future. For instructions see: http://journals.plos.org/plosone/s/submission-guidelines#loc-laboratory-protocols

We look forward to receiving your revised manuscript.

Kind regards,

Maria Gasset, Ph.D.

Academic Editor

PLOS ONE

Journal Requirements:

2. In your Methods section, please provide additional location information of the collection site, including geographic coordinates for the data set if available.

3. In your Methods section, please provide additional information regarding the permits you obtained for the work. Please ensure you have included the full name of the authority that approved the collection site access and, if no permits were required, a brief statement explaining why.

Reviewers' comments:

Reviewer's Responses to Questions

**Comments to the Author**

1. Is the manuscript technically sound, and do the data support the conclusions?

Reviewer #1: Yes

2. Has the statistical analysis been performed appropriately and rigorously? 

Reviewer #1: Yes

3. Have the authors made all data underlying the findings in their manuscript fully available?

Reviewer #1: Yes

4. Is the manuscript presented in an intelligible fashion and written in standard English?

Reviewer #1: Yes

5. Review Comments to the Author

Reviewer #1: The manuscript by Sharker and colleagues describes the identification and functional characterization of a prohormone convertase from the pleuropedal ganglion of the abalone Haliotis discus hannai. Although most of the analyses have been carried out quite efficiently, there are several points of this manuscript that require attention. I feel like the text could be improved by removing the unnecessary section about the structural modeling (it is unclear why this was performed by the authors since the results have not been discussed) and removing the parts concerning semi-quantitative PCR (which was just used as a preliminary analysis for qRT-PCR). Please find my detailed report attached below.

Line 143: This section is largely deficient in terms of the data provided. First, how were such sequences selected? Were all the sequences available in public databases used? Why were two different phylogenetic inference mathods (NJ and ME) used? In figure 3, the authors only reported the NJ tree, so it is needless to include the mention of the ME tree here.

Lines 149-150: too little information has been provided. It is important to specify which were the top scoring templates used for modeling (i.e. are other similar 3D structure available in the PDB database?).

Lines 171-172: “The melting temperature was 95°C for 10 s, 65°C for 1 min, and 97°C for 1 s as a default setting.”. This sentence seems out of place, as the cycling conditions have been described above. What are the authors referring to? I guess this sentence was supposed to describe the melting curve analysis, but its description is completely wrong.

Line 217: “This is the Fig 1 title.” -> ???

Line 226: “In silico analysis indicated that the deduced protein might be localized in the plasma membrane.” -> This is not very clear, considering that the following sentences do not describe any transmembrane domain. Which in silico analyses specifically indicate a plasma membrane localization? Based on literature data, what would be the expected subcellular localization of PC2? Is it common to find PC2 associated with the plasma membrane?

Line 256: “monophyletc outgroup”? What do the authors mean by this? This should be simply defined as “outgroup”.

Lines 265-266: “with a C-score between 2 and 4” -> NO. The C-score ranges from -4 and 2 and a given model is only characterized by a unique c-score, not by a range.

Lines 267-270: several critical information are missing here. Reporting proQ and ERRAT scores is quite useless if the authors do not report which structural templates were used for modeling. More in general, it looks like the structural modeling part had little sense in this manuscript, as it was not discussed in any meaningful way. My impression is that the authors were not very expert in the use of I-TASSER (which is a very easy-to-use online tool) and that their ability to discuss the outcome of the modeling process was very limited. Since the 3D structure of PC2 is not discussed in any way in this manuscript and the authors obviously have little idea about what they are doing here, I suggest that this part should be entirely removed, since it is pointless for the aims and scopes of this manuscript. The discussion found in lines 342-344 is very poor and uninformative. What was the reason to include structural modeling data in this manuscript is this is actually completely irrelevant for the discussion of the findings?

The semi-quantitative PCR analysis can be removed from the text. Obviously, this was just performed as a preliminary analysis to set-up the q-PCR analysis, which is a much superior method. Please remove this part (from the materials and methods and from the results section) and only leave the qPCR results.

Line 348: this should be expanded. What types of hormones are usually produced by this organ? Note that most of the readers of PlosONE are not familiar with abalone anatomy.

Line 546: CLUSTALW -> wasn’ ClustalOmega used (see materials and methods)

6. PLOS authors have the option to publish the peer review history of their article (what does this mean?). If published, this will include your full peer review and any attached files.

Reviewer #1: No

---

## [Author Response · Author response to Decision Letter 0]

13 Mar 2020

Reviewer #1: The manuscript by Sharker and colleagues describes the identification and functional characterization of a prohormone convertase from the pleuropedal ganglion of the abalone Haliotis discus hannai. Although most of the analyses have been carried out quite efficiently, there are several points of this manuscript that require attention. I feel like the text could be improved by removing the unnecessary section about the structural modeling (it is unclear why this was performed by the authors since the results have not been discussed) and removing the parts concerning semi-quantitative PCR (which was just used as a preliminary analysis for qRT-PCR). Please find my detailed report attached below.

Line 143: This section is largely deficient in terms of the data provided. First, how were such sequences selected? Were all the sequences available in public databases used? Why were two different phylogenetic inference mathods (NJ and ME) used? In figure 3, the authors only reported the NJ tree, so it is needless to include the mention of the ME tree here.

Response: We thank the reviewer for pointing this out and we agree with the reviewer. Therefore, we have added such information. To construct a phylogenetic tree, PC2 protein sequences from invertebrates and vertebrates were retrieved from the NCBI database using the BLASTP algorithm. This sentence is incorporated into lines151-152. Based on your valuable suggestion, I omitted the minimum evolution method from the line number 154.

Lines 149-150: too little information has been provided. It is important to specify which were the top scoring templates used for modeling (i.e. are other similar 3D structure available in the PDB database?).

Response: We thank the reviewer for pointing this out and we agree with the reviewer. Therefore, we have revised it accordingly and omitted from the abstract (line number 38-40) materials and methods (line number 155-160), and result (line number 266-275)section.

Lines 171-172: “The melting temperature was 95°C for 10 s, 65°C for 1 min, and 97°C for 1 s as a default setting.”. This sentence seems out of place, as the cycling conditions have been described above. What are the authors referring to? I guess this sentence was supposed to describe the melting curve analysis, but its description is completely wrong.

Response: We thank the reviewer for pointing this out and we agree with the reviewer. Therefore we revised it and omitted this sentence from the line number 182.

Line 217: “This is the Fig 1 title.” -> ???

Response: Based on the journal format, I incorporated the Fig. legend. 

Line 226: “In silico analysis indicated that the deduced protein might be localized in the plasma membrane.” -> This is not very clear, considering that the following sentences do not describe any transmembrane domain. Which in silico analyses specifically indicate a plasma membrane localization? Based on literature data, what would be the expected subcellular localization of PC2? Is it common to find PC2 associated with the plasma membrane?

Response: We thank the reviewer for pointing this out and we agree with the reviewer. Therefore, we have added such information. In silico analysis (protcomp, http://linux1.softberry.com/berry.phtml) indicated that the sub-cellular localization of this deduced protein is in the plasma membrane. This sentence is incorporated into lines 233-234. 

Line 256: “monophyletc outgroup”? What do the authors mean by this? This should be simply defined as “outgroup”.

Response: We thank the reviewer for pointing this out and we agree with the reviewer. Therefore, we revised it accordingly as shown in line number 261.

Lines 265-266: “with a C-score between 2 and 4” -> NO. The C-score ranges from -4 and 2 and a given model is only characterized by a unique c-score, not by a range.

Lines 267-270: several critical information are missing here. Reporting proQ and ERRAT scores is quite useless if the authors do not report which structural templates were used for modeling. More in general, it looks like the structural modeling part had little sense in this manuscript, as it was not discussed in any meaningful way. My impression is that the authors were not very expert in the use of I-TASSER (which is a very easy-to-use online tool) and that their ability to discuss the outcome of the modeling process was very limited. Since the 3D structure of PC2 is not discussed in any way in this manuscript and the authors obviously have little idea about what they are doing here, I suggest that this part should be entirely removed, since it is pointless for the aims and scopes of this manuscript. The discussion found in lines 342-344 is very poor and uninformative. What was the reason to include structural modeling data in this manuscript is this is actually completely irrelevant for the discussion of the findings? The semi-quantitative PCR analysis can be removed from the text. Obviously, this was just performed as a preliminary analysis to set-up the q-PCR analysis, which is a much superior method. Please remove this part (from the materials and methods and from the results section) and only leave the qPCR results.

Response: Based on your valuable suggestion, I omitted the 3D structure of PC2 and semi quantitative PCR from materials and methods, results, and discussion section. I also incorporated the following sentence in the line number 289-291.

qPCR assay was performed to investigate the mRNA expression profile of Hdh PC2 in neural ganglia (cerebral ganglia, branchial ganglia, pleuropedal ganglia), digestive gland, gonad (testis, ovary), gill, and mantle using gene-specific and ribosomal protein L-5 (RPL-5) primers.

Line 348: this should be expanded. What types of hormones are usually produced by this organ? Note that most of the readers of PlosONE are not familiar with abalone anatomy.

Response: We thank the reviewer for pointing this out and we agree with the reviewer. Please see the following sentences

Neuroendocrine hormone and their receptor such as GnRH (Kim et al., 2017), GnRH receptor (Sharker et al., unpublished), serotonin receptor (Sharker et al., 2020) and also steroid enzymes 17β HSD-11 (Sharker et al., unpublished) are synthesized from pleuropedal ganglion. PC2 (Neuroendocrine/prohormone convertase) seems to be involved in the posttranslational process of neuroendocrine hormone.

Line 546: CLUSTALW -> wasn’ ClustalOmega used (see materials and methods)

Response: We thank the reviewer for pointing this out and we agree with the reviewer. Therefore we have inserted it as shown in line number 548.

Journal Requirements:

Response: The manuscript style meets the PLOS ONE's style requirements.

2. In your Methods section, please provide additional location information of the collection site, including geographic coordinates for the data set if available.

Response: Based on your valuable suggestion, I incorporated the location of collection site in the line number 86.

3. In your Methods section, please provide additional information regarding the permits you obtained for the work. Please ensure you have included the full name of the authority that approved the collection site access and, if no permits were required, a brief statement explaining why.

Response: Based on your valuable suggestion, we have added such information in the line number 95-101.

Animal experiments were conducted in accordance with the guidelines of the Institutional Animal Care and Use Committee of Chonnam National University (CNU IACUC) and according to Article 14th of the Korean Animal Protection Law of the Korean government, and the animals were cared for in accordance with the Guidelines for Animal Experiments of Chonnam National University. No specific permissions are required to work with invertebrates in South Korea. Similarly, no permissions were needed for the collection of H. discus hannai from sample sites because they were not harvested from the protected area and this species is not an endangered or protected species.

 Ethical approval information of abalone

All vertebrate animal research requires approval from the institution's animal care and usage committee. However, this is not necessarily for invertebrates. Abalone (Haliotis discus hannai) is an invertebrate gastropod mollusk. So, abalone does not require approval from the authority. 

Response: I incorporated the gel figure as supporting information.

---

## [Decision Letter · Decision Letter 1]

18 Mar 2020

PONE-D-20-03317R1

Molecular characterization and spatiotemporal expression of prohormone convertase 2 in the Pacific abalone, Haliotis discus hannai

PLOS ONE

Dear Dr Kang Hee Kho,

Thank you for submitting your revised manuscript to PLOS ONE. After careful consideration, we feel that it still needs a few minor corrections. Therefore, we invite you to submit a revised version of the manuscript that addresses the points raised during the review process.

We would appreciate receiving your revised manuscript by May 02 2020 11:59PM. To enhance the reproducibility of your results, we recommend that if applicable you deposit your laboratory protocols in protocols.io, where a protocol can be assigned its own identifier (DOI) such that it can be cited independently in the future. For instructions see: http://journals.plos.org/plosone/s/submission-guidelines#loc-laboratory-protocols

We look forward to receiving your revised manuscript.

Kind regards,

Maria Gasset, Ph.D.

Academic Editor

PLOS ONE

Reviewers' comments:

Reviewer's Responses to Questions

**Comments to the Author**

1. If the authors have adequately addressed your comments raised in a previous round of review and you feel that this manuscript is now acceptable for publication, you may indicate that here to bypass the “Comments to the Author” section, enter your conflict of interest statement in the “Confidential to Editor” section, and submit your "Accept" recommendation.

Reviewer #1: (No Response)

2. Is the manuscript technically sound, and do the data support the conclusions?

Reviewer #1: Yes

3. Has the statistical analysis been performed appropriately and rigorously? 

Reviewer #1: Yes

4. Have the authors made all data underlying the findings in their manuscript fully available?

Reviewer #1: Yes

5. Is the manuscript presented in an intelligible fashion and written in standard English?

Reviewer #1: Yes

6. Review Comments to the Author

Reviewer #1: Thank you for providing a revised version of this manuscript. I feel like the manuscript has been significantly improved with the removal of the 3D modeling part and a better definition of some key passages of the main text.

There are however some important points that still require attention.

1) The authors mention the use of Softberry to predict the subcellular localization of this protein, but provided a link to an outdated version of this portal. The authors should state that the tools used for this prediction is ProtComp 9.0 and provide the appropriate reference. Moreover, it appears that the subcellular localization of the protein is not the plasma membrane, but the Golgi apparatus. I have retrieved the sequence myself using the accession ID provided by the authors and verified that the predicted subcellular localization is Golgi. This would make much more sense than having a transmembrane localization, as proteolytic processing often happens in the Golgi network (see for example https://doi.org/10.1074/jbc.M507193200. I would urge the authors to correct this result and add necessary literature references to works that report the expected subcellular localization of similar proteases.

2) The authors updated their phylogenetic tree and mentioned that the sequences added were detected through BLASTp. Which selection criteria were used? Any specific threshold of primary sequence homology? The authors should explain whether other paralogous genes were present and how these were excluded.

7. PLOS authors have the option to publish the peer review history of their article (what does this mean?). If published, this will include your full peer review and any attached files.

Reviewer #1: Yes: Marco Gerdol

---

## [Author Response · Author response to Decision Letter 1]

19 Mar 2020

Reviewer #1: Thank you for providing a revised version of this manuscript. I feel like the manuscript has been significantly improved with the removal of the 3D modeling part and a better definition of some key passages of the main text.

There are however some important points that still require attention.

1) The authors mention the use of Softberry to predict the subcellular localization of this protein, but provided a link to an outdated version of this portal. The authors should state that the tools used for this prediction is ProtComp 9.0 and provide the appropriate reference. Moreover, it appears that the subcellular localization of the protein is not the plasma membrane, but the Golgi apparatus. I have retrieved the sequence myself using the accession ID provided by the authors and verified that the predicted subcellular localization is Golgi. This would make much more sense than having a transmembrane localization, as proteolytic processing often happens in the Golgi network (see for example https://doi.org/10.1074/jbc.M507193200. I would urge the authors to correct this result and add necessary literature references to works that report the expected subcellular localization of similar proteases.

Response: We thank the reviewer for pointing this out and we agree with the reviewer. Therefore, we have added such information. In silico analysis (protcomp, http://www.softberry.com/berry.phtml) indicated that the sub-cellular localization of this deduced protein is in the membrane bound golgi network. This sentence is incorporated into lines 216-217. We have also added ‘The predicted protein might be localized in the membrane bound golgi network which is in agreement with the results of previous studies [6]’ in the line number 306-307.

2) The authors updated their phylogenetic tree and mentioned that the sequences added were detected through BLASTp. Which selection criteria were used? Any specific threshold of primary sequence homology? The authors should explain whether other paralogous genes were present and how these were excluded. 

Response: Thank you very much for your valuable comments. We did not consider any specific threshold for selecting sequence homology. Based on your valuable suggestions, we included paralogous genes of PCs and reconstructed the phylogenetic tree. We have added following information in the line numbers 241-243 and inserted the reconstructed phylogenetic tree figure (Fig. 3)

‘The constructed phylogenetic tree revealed several distinct clades. Hdh PC2 is contained in the gastropod PC2 clade and is more closely related to Has PC2’.

---

## [Decision Letter · Decision Letter 2]

23 Mar 2020

Molecular characterization and spatiotemporal expression of prohormone convertase 2 in the Pacific abalone, Haliotis discus hannai

PONE-D-20-03317R2

Dear Dr. Kang Hee Kho,

We are pleased to inform you that your manuscript has been judged scientifically suitable for publication and will be formally accepted for publication once it complies with all outstanding technical requirements.

With kind regards,

Maria Gasset, Ph.D.

Academic Editor

PLOS ONE

Additional Editor Comments (optional):

Reviewers' comments:

Reviewer's Responses to Questions

**Comments to the Author**

1. If the authors have adequately addressed your comments raised in a previous round of review and you feel that this manuscript is now acceptable for publication, you may indicate that here to bypass the “Comments to the Author” section, enter your conflict of interest statement in the “Confidential to Editor” section, and submit your "Accept" recommendation.

Reviewer #1: All comments have been addressed

2. Is the manuscript technically sound, and do the data support the conclusions?

Reviewer #1: Yes

3. Has the statistical analysis been performed appropriately and rigorously? 

Reviewer #1: Yes

4. Have the authors made all data underlying the findings in their manuscript fully available?

Reviewer #1: Yes

5. Is the manuscript presented in an intelligible fashion and written in standard English?

Reviewer #1: Yes

6. Review Comments to the Author

Reviewer #1: Thank. All my concerns have been appropriately addressed. In my opinion the manuscript is now accepyable for publication.

7. PLOS authors have the option to publish the peer review history of their article (what does this mean?). If published, this will include your full peer review and any attached files.

Reviewer #1: No

---

## [Editor Report · Acceptance letter]

27 Mar 2020

PONE-D-20-03317R2 

Molecular characterization and spatiotemporal expression of prohormone convertase 2 in the Pacific abalone, *Haliotis discus hannai*

Dear Dr. Kho:

I am pleased to inform you that your manuscript has been deemed suitable for publication in PLOS ONE. Congratulations! Your manuscript is now with our production department. 

With kind regards,

on behalf of

Dr. Maria Gasset 

Academic Editor

PLOS ONE